# pyTFM: A tool for traction force and monolayer stress microscopy

**Andreas Bauer**[1]*, **Magdalena Prechová**[2], **Lena Fischer**[1], **Ingo Thievessen**[1], **Martin Gregor**[2], **Ben Fabry**[1]

**1** Department of Physics, Friedrich-Alexander University Erlangen-Nürnberg, Erlangen, Germany,
**2** Laboratory of Integrative Biology, Institute of Molecular Genetics of the Czech Academy of Sciences, Prague, Czech Republic

\* andreas.b.bauer@fau.de

## Abstract

Cellular force generation and force transmission are of fundamental importance for numerous biological processes and can be studied with the methods of Traction Force Microscopy (TFM) and Monolayer Stress Microscopy. Traction Force Microscopy and Monolayer Stress Microscopy solve the inverse problem of reconstructing cell-matrix tractions and inter- and intra-cellular stresses from the measured cell force-induced deformations of an adhesive substrate with known elasticity. Although several laboratories have developed software for Traction Force Microscopy and Monolayer Stress Microscopy computations, there is currently no software package available that allows non-expert users to perform a full evaluation of such experiments. Here we present pyTFM, a tool to perform Traction Force Microscopy and Monolayer Stress Microscopy on cell patches and cell layers grown in a 2-dimensional environment. pyTFM was optimized for ease-of-use; it is open-source and well documented (hosted at https://pytfm.readthedocs.io/) including usage examples and explanations of the theoretical background. pyTFM can be used as a standalone Python package or as an add-on to the image annotation tool *ClickPoints*. In combination with the *ClickPoints* environment, pyTFM allows the user to set all necessary analysis parameters, select regions of interest, examine the input data and intermediary results, and calculate a wide range of parameters describing forces, stresses, and their distribution. In this work, we also thoroughly analyze the accuracy and performance of the Traction Force Microscopy and Monolayer Stress Microscopy algorithms of pyTFM using synthetic and experimental data from epithelial cell patches.

## Author summary

The analysis of cellular force generation and transmission is an increasingly important aspect in the field of biological research. However, most methods for studying cellular force generation or transmission require complex calculations and have not yet been implemented in comprehensive, easy-to-use software. This is a major hurdle preventing a wider application in the field. Here we present pyTFM, an open-source Python package with a graphical user interface that can be used to evaluate cellular force generation in

**Data Availability Statement:** Raw images of cells and cell substrate in tensed and relaxed state underlying the results presented in the study are available from the Zenodo database. DOI: 10.5281/zenodo.4047040; URL: https://zenodo.org/record/4047040 The data points plotted in Fig 4 are

provided in the S1 Dataset. The source code is available in the S1 Archive file.

**Funding:** This work was funded by the Deutsche Forschungsgemeinschaft (SFB-TRR 225, project number 326998133, subprojects A01 (A.B., B.F.), B06 (L.F., I.T.), and FA 336/11-1 (B.F.)), the Ministry of Health of the Czech Republic (grant 17-31538A) (M.P., M.G.) and The European Cooperation in Science and Technology (COST) grant CA15214-EuroCellNet (MEYS CR LTC17063) (M.P., M.G.). The funders had no role in study design, data collection and analysis, decision to publish, or preparation of the manuscript.

**Competing interests:** The authors have declared that no competing interests exist.

cells and cell colonies and force transfer within small cell patches and larger cell layers grown on the surface of an elastic substrate. In combination with the image annotation and tool *ClickPoints*, pyTFM allows the user to set all necessary analysis parameters, select regions of interest, examine the input data and intermediary results, and calculate a wide range of parameters describing cellular forces, stresses, and their distribution. Additionally, pyTFM can be used as standalone python library. pyTFM comes with an extensive documentation (hosted at https://pytfm.readthedocs.io/) including usage examples and explanations of the theoretical background.

This is a *PLOS Computational Biology* Software paper.

## 1 Introduction

The generation of active forces gives cells the ability to sense the mechanical properties of their surroundings [1], which in turn can determine the cell fate during differentiation processes [2], the migratory behavior of cells [3] or the response to drugs [4]. Measuring cellular force generation is important for understanding fundamental biological processes including wound healing [5], tissue development [6], metastasis formation [7, 8] and cell migration [3].

Cellular forces can be divided into three categories: Forces that are transmitted between a cell and its surrounding matrix (also referred to as traction forces), forces that are transmitted between cells, and forces that are transmitted inside cells.

Traction forces can be measured with Traction Force Microscopy (TFM), which is most easily applied to cells grown in a 2-dimensional environment: Cells are seeded on a planar elastic substrate on which they adhere, spread, and exert forces. The substrate contains fiducial markers such as fluorescent beads for tracking cell force-induced deformations of the substrate. Typically, the substrate is imaged in a tensed and a relaxed (force-free) state, whereby force relaxation is achieved by detaching the cells from the substrate. These two images are then compared to quantify substrate deformations, either by Particle Tracking Velocimetry (PTV) where individual marker beads are tracked, or by cross-correlation based Particle Image Velocimetry (PIV) [9]. The deformation field of the substrate is subsequently analyzed to calculate the cell-generated tractions in x- and y-directions. (Note that if the substrate deformations in z-direction are also measured, which requires at least one additional image taken at a different focal plane, it is possible to compute the tractions in z-direction [10]. In what follows, however, we ignore deformations and tractions in z-direction).

The calculation of the traction field from the deformation field is an inverse problem for which a number of algorithms have been developed, including numerical methods such as the Boundary Elements Method [11, 12], Fourier-based deconvolution [13], and Finite Element (FE) computations [14], all of which have specific advantages and disadvantages (see [15] and [12] for a detailed discussion). pyTFM uses the Fourier Transform Traction Cytometry (FTTC) algorithm [13], as it is computationally fast and does not require knowledge of the cell boundary.

Tractions must be balanced by forces transmitted within or between cells. These forces are usually described by stress tensors. The stress tensor field for cells grown in a 2-dimensional environment can be calculated using the Monolayer Stress Microscopy method [16, 17], whereby the cell or cell patch is modeled as an elastically stretched 2-dimensional sheet in contact with the matrix so that the external tractions are balanced by the internal stress of the elastic sheet.

In pyTFM, the cell or cell patch is modelled as a linear elastic sheet represented by a network of nodes and vertices so that the stresses can be calculated by a standard two-dimensional Finite Element Method (FEM). First, forces with the same magnitude but opposing direction to the local tractions are applied to each node. Then, internal strains and consequently stresses are calculated based on the network geometry and elastic properties.

pyTFM uses the Monolayer Stress Microscopy algorithm developed by Tambe et al. 2013 [17]. In this implementation, the calculated network strain has no physical meaning, as the matrix strain and the cell strain are not required to match [18]. Consequently, in the limit of homogeneous elastic properties throughout the cell sheet, its Young's modulus has no influence on the stress estimation, and its Poisson's ratio has only a negligible influence. Both parameters can therefore be freely chosen [17]. Note that there are different implementations of Monolayer Stress Microscopy in which cell and matrix deformations are coupled and the network elasticity corresponds to the effective cell elasticity, which must be known to obtain correct results [19]. A comparison about these two approaches can be found in [18].

For the calculation of monolayer stresses of small cell patches, pyTFM corrects errors caused by a low spatial resolution of Traction Force Microscopy. In the case of low resolution, a significant part of the tractions can seem to originate from outside the cell area, and when only tractions beneath the cell area are considered, the stress field is underestimated. This problem cannot be remedied by constraining the tractions to be zero outside the cell area (constrained TFM) as this tends to produce large spurious tractions at the cell perimeter [13] and hence unphysically high stresses in the cell monolayer. Ng et al. 2014 [20] addressed this issue by expanding the FEM-grid to cover all tractions generated by the cell patch and by exponentially decreasing the stiffness of the FEM-grid with increasing distance to the cell patch edge. In our implementation, the FEM-grid is also expanded to cover all cell-generated tractions, however, we found it unnecessary to introduce a stiffness gradient in the FEM-grid. Moreover, zero-translation and zero-rotation constraints are explicitly added to the FEM-algorithm in pyTFM.

Finally, pyTFM adds a number user-friendly features to easily set parameters, select regions of interest and quickly evaluate results. For this, pyTFM can be optionally used as an add-on to the image annotation tool *ClickPoints* [21]. This makes the analysis of large data sets particularly easy by sorting input and output data in a database and allowing the user to browse through it.

pyTFM is well documented, including detailed usage examples, information on the theory of TFM and Monolayer Stress Microscopy, and explanations about the calculated parameters. The documentation is hosted at https://pytfm.readthedocs.io.

## 2 Design and implementation

pyTFM is a Python package implemented in Python 3.6. It is mainly intended to be used as an add-on for the image display and annotation tool *ClickPoints*, but can also be used as a stand-alone Python library.

pyTFM performs TFM and Monolayer Stress Microscopy following the workflow shown in Fig 1A. The main steps of the workflow are the calculation of the deformation field from images of the cell substrate in a tensed and relaxed state, the calculation of the traction field, and the calculation of the monolayer stress field. The mathematical details of these steps are discussed in Section 2.2. Deformation, traction and stress fields are further analyzed to extract scalar measures of cellular stress, force generation, and force transmission between cells.

Cellular force generation is quantified by the total force generation and centripetal contractility. Total force generation in turn is described by the strain energy that is elastically stored in the substrate, and centripetal contractility is described by the sum of all cell-generated forces projected towards a single force epicenter. Stresses are quantified by average normal and shear

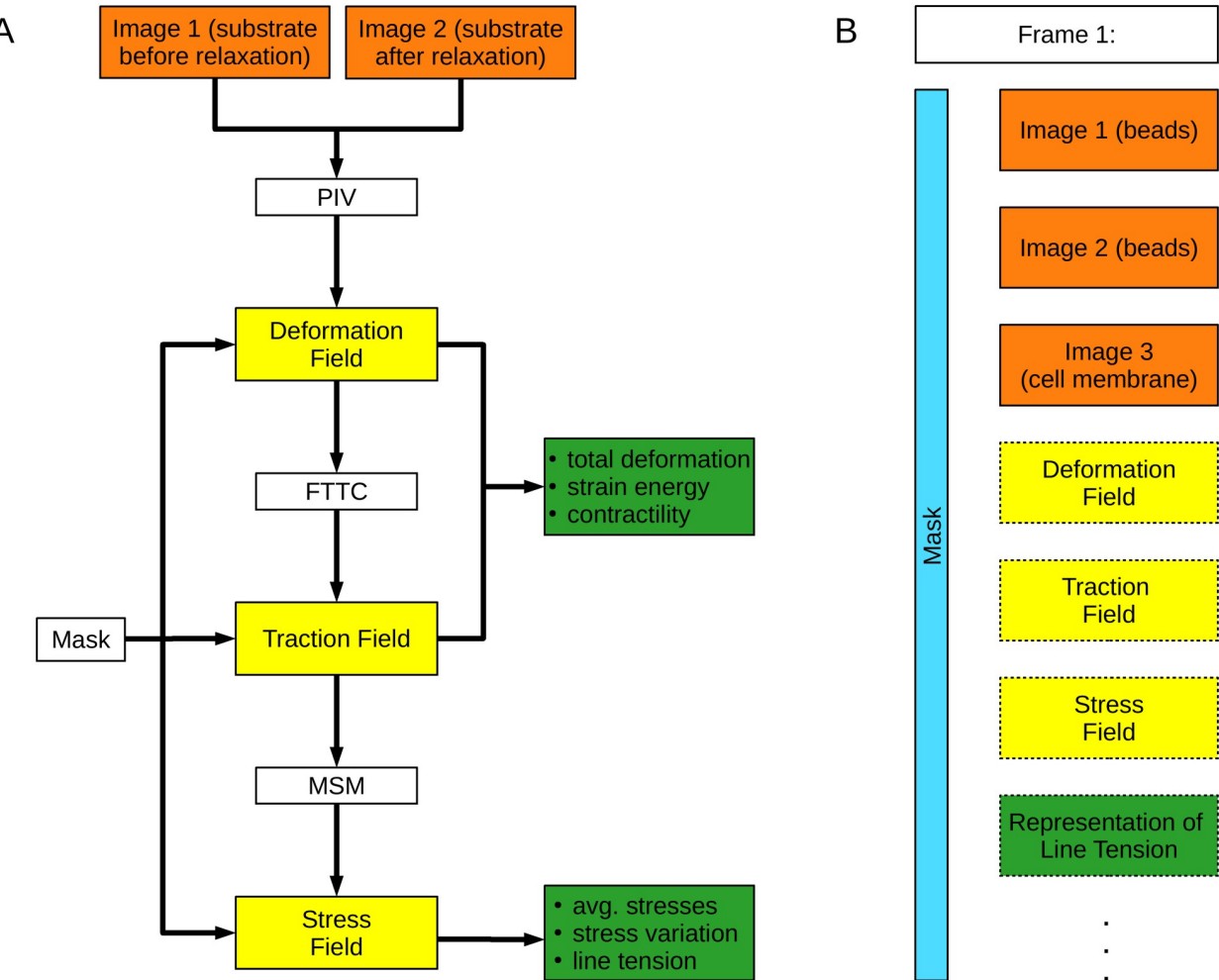

**Fig 1. Workflow of pyTFM and image database organization.** A: Workflow of TFM and Monolayer Stress Microscopy analysis with pyTFM. B: Organization of the pyTFM *ClickPoints* database. Input images are colored in orange, intermediary results in yellow, and the final output in the form of scalar measures in green. The mask that defines the cell boundaries and the area over which strain energy, contractility and monolayer stresses are computed is colored light blue.

stresses and their coefficient of variation, which is a measure for stress fluctuations. Cell-cell force transmission is quantified by the line tension, which is the force per unit length acting on a segment of a cell-cell boundary. Specifically, pyTFM calculates the average magnitude of the line tension as well as the average normal and shear component of the line tension. Additionally, pyTFM calculates the area and number of cells of each cell patch, which can be used to normalize the quantities above. We provide more details on how these quantities are defined and how to interpret them in Supplementary S1 Text.

The user is required to select an area of the traction field over which the strain energy, contractility and monolayer stresses are computed. This area should cover all cell-generated tractions and is thus typically larger than the cell area. However, a significant further extension of the user-selected area beyond the cell edge will lead to an underestimation of monolayer stresses, as will be further discussed in Section 2.2.2. Optionally, the outline of the cell or cell patch can be selected, defining the area over which average stresses and stress fluctuations are

computed. Also optionally, the outline of cell-cell boundaries can be selected to calculate force transmission between cells.

pyTFM generates several output files. All fields (deformations, tractions, stresses) are saved in the form of NumPy arrays as binary.npy files and are plotted as vector fields or heat maps. The cell-cell force transmission and the strain energy density can also be plotted (see Section 3.2 for an example). The user has full control over which plots are produced. All calculated scalar results are saved in a tab-separated text file. pyTFM includes Python functions to read, compare and statistically analyze the result text files of several experiments. Alternatively, the result text files can be opened with standard text editors or data analysis tools such as Excel.

pyTFM builds on a number of well established open source python packages. Most notably, *numpy* [22], *cython* [23] and *scipy* [24] for a variety of computational tasks, *scikit-image* [25] for the automated detection of cell boundaries, estimation of cell number and microscope stage drift correction and *matplotlib* [26] for the display of vector fields and cell-cell forces.

## 2.1 Integration of pyTFM with *ClickPoints* databases

When using the pyTFM add-on in *ClickPoints*, input and output images are organized in a database (Fig 1B), which allows users to efficiently navigate large data sets. The database is organized in frames and layers: Each frame represents one field of view. Initially, three layers are assigned to each frame. These layers contain images of the substrate in the tensed and relaxed state, and an image of the cells. Output plots such as the deformation field or the traction field are added as new layers in each analysis step. Additionally, each frame is associated with a mask object in the form of an integer array representing the user selected areas and cell outlines. This mask object can be drawn directly in *ClickPoints* and can be displayed in each layer of a frame.

pyTFM provides a graphical user interface for the *ClickPoints* environment, which allows the user to select input images, to set all relevant analysis parameters (e.g. the elasticity of the substrate), and to select whether the analysis should be performed on all frames or just the currently viewed frame (Fig 2). A number of tools are provided by *ClickPoints*, e.g. to draw masks, to adjust contrast and brightness of the displayed images, to measure distances and object sizes, and to export images and video sequences.

## 2.2 Implementation of TFM and monolayer stress microscopy

**2.2.1 Deformation fields and TFM.** Deformation fields are calculated from the images of the substrate in a tensed and relaxed state using the cross correlation-based Particle Image Velocimetry (PIV) algorithm implemented in the openPIV Python package [9]. PIV is performed by selecting for example a 50x50 pixel tile around a given pixel from the tensed image and shifting the tile by pixel increments in all directions across the corresponding tile in the relaxed image. This yields a correlation matrix of in this case 99x99 pixels. The deformation vector is then obtained by calculating the vector between the position of the highest correlation and the center of the matrix. The initial deformation vector is further refined to sub-pixel accurate values by fitting a 2D Gauss curve to the directly neighbouring correlation values. To reduce noise, deformation vectors with a signal-to-noise ratio smaller than 1.03 are exclude and replaced by the local mean of the surrounding deformations at distances $< = 2$ pixel. The signal-to-noise ratio of each deformation vector is defined as the ratio of the correlation of the highest peak and the correlation of the second-highest peak outside of a neighborhood of 2 pixels around the highest peak.

A common issue when calculating the deformation field is a global drift of the images, which needs to be corrected prior to PIV. This drift can reach several μm and is caused by the

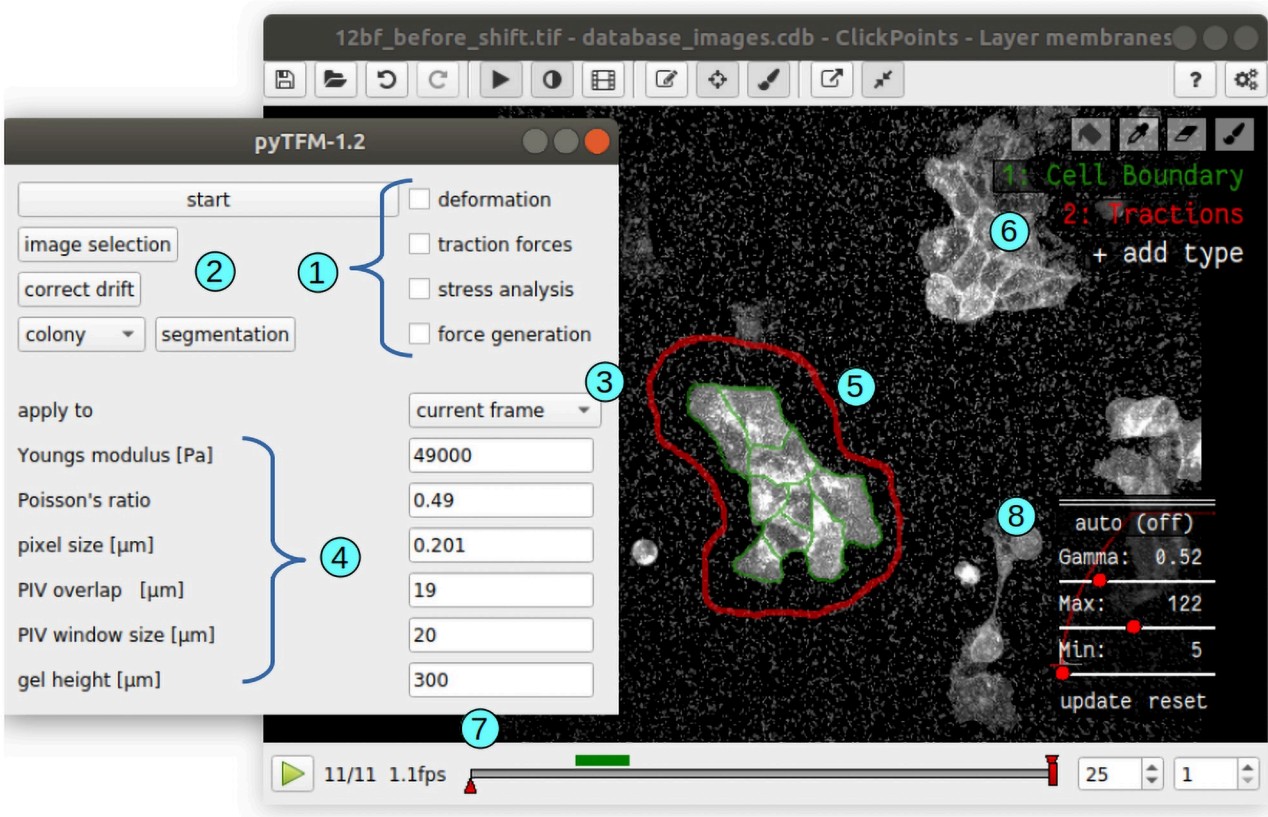

**Fig 2. User interface of pyTFM.** 1: Check boxes to select specific analysis steps. 2: Selection of input images, drift correction and semi automatic segmentation of cell borders. 3: Drop-down menu to select between analysing all frames in a database or analysing only the currently viewed frame. 4: Parameters for PIV and TFM. 5: User-selected region (red outline) and cell boundaries (green) for computing tractions, stresses, contractility, strain energy and line tensions. 6: *ClickPoints* tools to select the region and the cell boundaries by drawing masks. 7: *ClickPoints* navigation bar through frames. Layers are navigated with the Page Up and Page Down keys, and frames are navigated with the left and right arrow keys. 8: *ClickPoints* panel to adjust contrast and brightness of the image display. This is helpful for manually segmenting cell borders.

combined effect of positioning inaccuracies of the motorized x, y-stage, mechanical handling or shaking during the addition of trypsin-EDTA, and slow temporal or temperature-induced mechanical drift. pyTFM offers a global drift correction that works in three steps: First, the drift is estimated with sub-pixel accuracy by cross-correlating the entire first image with the entire second image. This is done with the "phase_cross_correlation" function from the *scikit-image* [25] python package. Next, the first image is shifted by the drift, and finally both images are cropped to the overlapping field of view.

Tractions are calculated with the Fourier Transform Traction Cytometry (FTTC) method [13]. Deformations ($\vec{u}$) and tractions ($\vec{t}$) are related by the convolution of the traction vector field and a Greens tensor $K$:

$$\vec{u} = K \otimes \vec{t} \tag{1}$$

In the case of a linearly elastic semi-infinite substrate, $K$ is given by the Boussinesq equations [27]. Inverting Eq 1 and solving for the tractions is difficult in real space. However, by exploiting the convolution theorem, the equation simplifies to a multiplication in Fourier

space:

$$\tilde{\vec{u}}(\vec{k}) = \tilde{K}(\vec{k})\tilde{\vec{T}}(\vec{k}) \tag{2}$$

where $\tilde{\vec{u}}(\vec{k})$, $\tilde{\vec{T}}(\vec{k})$ and $\tilde{K}(\vec{k})$ are the Fourier transforms of the deformation field, the traction field and the Greens tensor. The latter can be found in [13].

Eq 2 can be analytically solved and thus allows for the direct calculation of tractions in Fourier space. Tractions in real space are then obtained by applying the inverse Fourier transform.

One particular challenge of Traction Force Microscopy is that noise in the deformation field can lead to large errors in the traction field. This can be remedied by regularization of the reconstructed forces, e.g. by adding the L1 or L2 norm to the cost function of the inverse minimization problem [28]. By contrast, pyTFM does not use explicit regularization but instead smooths the calculated traction field with a user-defined Gaussian low-pass filter, with a sigma of typically 3 μm. This effectively suppresses all tractions with high spatial frequencies but inherently limits the spatial resolution of the pyTFM algorithm. The appropriate degree of smoothing depends on the spatial resolution of the deformation field, which in turn depends e.g. on the density of fiducial markers, the window size for the PIV algorithm or image noise. The user is encouraged to test different values for sigma and to select the smallest value for which the noise in the cell-free areas is still tolerable in comparison to the magnitude of cell tractions.

The original TFM algorithm assumes that the underlying substrate is infinitely thick, which is justified in the case of single cells with dimensions that are smaller than the thickness of the elastic substrate. In the case of cell patches, however, this assumption is inadequate. We have therefore included a correction term for finite substrate thickness [29].

**2.2.2 Monolayer stress microscopy.** Stresses in a cell sheet are calculated with an implementation of Monolayer Stress Microscopy as described in [16, 17]. For computing stresses in small cell patches, we implemented a method that corrects for the limited spatial resolution of unconstrained TFM, which otherwise would lead to a substantial underestimation of stresses [20]. Details of this correction are described below.

In the absence of inertial forces, tractions and stresses are balanced according to the relation:

$$-t_x = \frac{\delta\sigma_{xx}}{\delta x} + \frac{\delta\sigma_{yx}}{\delta y}$$
$$-t_y = \frac{\delta\sigma_{yx}}{\delta x} + \frac{\delta\sigma_{yy}}{\delta y} \tag{3}$$

where $\sigma_{xx}$, $\sigma_{yy}$ are the normal stresses in x- and y- direction, $\sigma_{yx}$ is the shear stress, and $t_x$ and $t_y$ are the x- and y-components of the traction vector. This differential equation is solved using a Finite Element method (FEM) where the cell patch is modeled as a 2-dimensional network of nodes arranged in a grid of quadrilateral elements. Each node in the FEM-grid is loaded with a force of the same magnitude but opposing direction as the local tractions. In the standard FE method, the nodal displacements $\vec{d}$ of the cell patch are calculated by solving the equation

$$\vec{d} = K^{-1}\vec{f} \tag{4}$$

where $\vec{f}$ are the vector of nodal forces, and $K^{-1}$ is the inverse stiffness matrix. The nodal displacements are converted to strains by taking the derivative in x- and y-direction. Then, the strain is used to calculate the stress from the stress-strain relationship of a linearly elastic

2-dimensional material:

$$
\begin{pmatrix} \sigma_{11} \\ \sigma_{22} \\ \sigma_{12} \end{pmatrix} = \frac{E}{1-\nu^2} \begin{pmatrix} 1 & \nu & 0 \\ \nu & 1 & 0 \\ 0 & 0 & 1-\nu \end{pmatrix} \begin{pmatrix} \epsilon_{11} \\ \epsilon_{22} \\ \epsilon_{12} \end{pmatrix}
\tag{5}
$$

where E and $\nu$ are Young's modulus and Poisson's ratio of the material, and $\epsilon_{11}$, $\epsilon_{22}$ and $\epsilon_{12}$ are the components of the strain tensor. Most of the FEM calculation is performed using the *solid-spy* Python package [30].

The stiffness matrix $K$ in Eq 4 depends on the Young's modulus in such a way that the Young's modulus in Eq 5 cancels out. The traction-stress relation is therefore independent of the Young's modulus of the cell patch [17]. Furthermore, the Poisson's ratio has only a negligible influence on the stress prediction [17]. In the pyTFM algorithm, the Young's modulus is set to 1 Pa, and the Poisson's ratio is set to 0.5.

The FEM algorithm assumes that there are no torques or net forces acting on the cell patch. This must also be true in reality as the cell patch would not be stationary otherwise. However, the TFM algorithm only ensures that forces and torques are globally balanced (across the entire image), but not necessarily across a cell patch. These unbalanced net forces and torques acting on a cell patch must be corrected prior to performing the FEM algorithm to accurately compute the cellular stresses. pyTFM corrects unbalanced net forces by subtracting the sum of all force vectors of the FEM system from the force vector at each node. The unbalanced net torque is corrected by rotating the direction of all force vectors by a small angle (typically below 5˚) until the torque of the entire system is zero.

By constraining the FEM system to zero rigid rotational or translational movement, Eq 4 is uniquely solvable [31]. These constraints can be applied in two ways: The first option is to apply a boundary condition of zero displacement in x- and y- direction to an arbitrarily chosen node of the FEM grid, and a boundary condition of zero rotation between this fixed node and another arbitrarily chosen second node [17]. In practise, this is implemented by selecting a second node with the same y-coordinate as the fixed node and applying a zero x-displacement boundary condition. The second option, which is implemented in pyTFM, is as follows: Instead of subjecting individual nodes to displacement boundary conditions, we formulate zero rigid displacement and rotation conditions on the whole system in three separate equations (Eqs 6–8), add them to the system of equations in Eq 4 and finally solve the combined system numerically using a standard least-squares minimization. Eqs 6 and 7 ensure that the sum of all nodal displacements in x- and y-direction is zero, and Eq 8 ensures that the rotation of all nodes around the center of mass of the FEM system is zero.

$$
\sum(d_x) = 0
\tag{6}
$$

$$
\sum(d_y) = 0
\tag{7}
$$

$$
\sum(d_x r_y - d_y r_x) = 0
\tag{8}
$$

$r_x$ and $r_y$ are the components of the distance vector of the corresponding node to the center of mass of the FEM-grid. Note that the FEM algorithm described above models the cell sheet as a linear elastic material. However, in the future, other FEM algorithms suitable for non-linear elastic materials could also be applied.

The analysis of stresses in small cell patches poses a second challenge: The FEM-grid should be of the same size and shape as the cell patch, as outside nodes add additional stiffness, leading

to an underestimation of the stress field. However, the limited spatial resolution of both PIV and TFM implies that some forces generated close to the edge of the cell patch are predicted to originate from outside the cell patch. Neglecting these forces would lead to an underestimation of the stress field. This can be avoided by extending the FEM-grid by a small margin so that all cell-generated forces are included in the analysis. In practice, the user outlines the area with clearly visible tractions (red outline in Fig 2), over which pyTFM then spans the FEM-grid. We explain further details of this approach in Section 3.1.1.

**2.2.3 Limits of applicability of monolayer stress microscopy and TFM.** The TFM and Monolayer Stress Microscopy algorithms can only be applied if a number of conditions are met. 2-dimensional TFM relies on the assumption that tractions in z-direction generate only small deformations in the x- and y-plane. This is valid if z-tractions are small, or if the substrate is almost incompressible (Poison's ratio close to 0.5) [11] Additionally, TFM assumes that the matrix is a linearly elastic material. Both assumptions are valid for polyacrylamide and PDMS, two popular substrates for TFM [32–35].

For Monolayer Stress Microscopy, cells are modeled as a linearly elastic material with homogeneous and isotropic elastic properties. Although many cell types have been shown to exhibit stress stiffening (the Young's modulus increases with the cellular stress [36]), this has only a small effect on the stresses recovered by Monolayer Stress Microscopy [17].

Furthermore, Monolayer Stress Microscopy models the cells as a 2-dimensional flat sheet. Deviations from this assumption can introduce an error to the stress calculation on the order of $(h/l)^2$ [17], where $h$ is the cell height and $l$ is the wavelength of the tractions in Fourier-space (in the case of a single cell that generates tractions in the form of two opposing force monopoles, $l$ corresponds to the distance between the force monopoles). This error can become relevant for isolated round cells but not for larger flat cell colonies.

# 3 Results

## 3.1 Accuracy of TFM and MSM algorithms

To evaluate the accuracy of the calculated tractions and stresses, we designed a simple test system with a predefined stress field for which tractions and deformations can be analytically computed. We then compare the analytical solution to the solution provided by pyTFM.

The workflow of this test is illustrated in Fig 3A: First, we define a square-shaped area of 150 μm width representing a cell patch. This area carries a uniform normal stress in x- and y-direction of 1 N/μm magnitude and zero shear stress. Stresses outside the cell patch are set to zero. Next, we calculate the corresponding traction field by taking the spatial derivatives of the stress field and applying Eq 3.

From the traction field, we obtain the deformation field by first calculating the Fourier transform of the traction field. Then we use Eq 2 to obtain the deformation field in Fourier space and, after applying the inverse Fourier transform, in real space. We use a modified Greens Tensor $K$ to account for a finite substrate thickness [29]. The substrate thickness is set to 100 μm.

The deformation field is then used as the input for the TFM and Monolayer Stress Microscopy algorithms. We use an FEM-grid area that is 5 μm larger than the original stress field area since this resulted in the best stress recovery (Fig 4A).

The computed mean of the normal and shear stresses and the standard deviation of the normal stresses are finally compared with the known input stress (uniform normal stress in x- and y-direction of 1N/μm magnitude, and zero shear stress). To compare the reconstructed traction field with the analytical solution, we also compute the total contractility (sum of all cell-generated forces projected towards a single force epicenter) over the FEM-grid area.

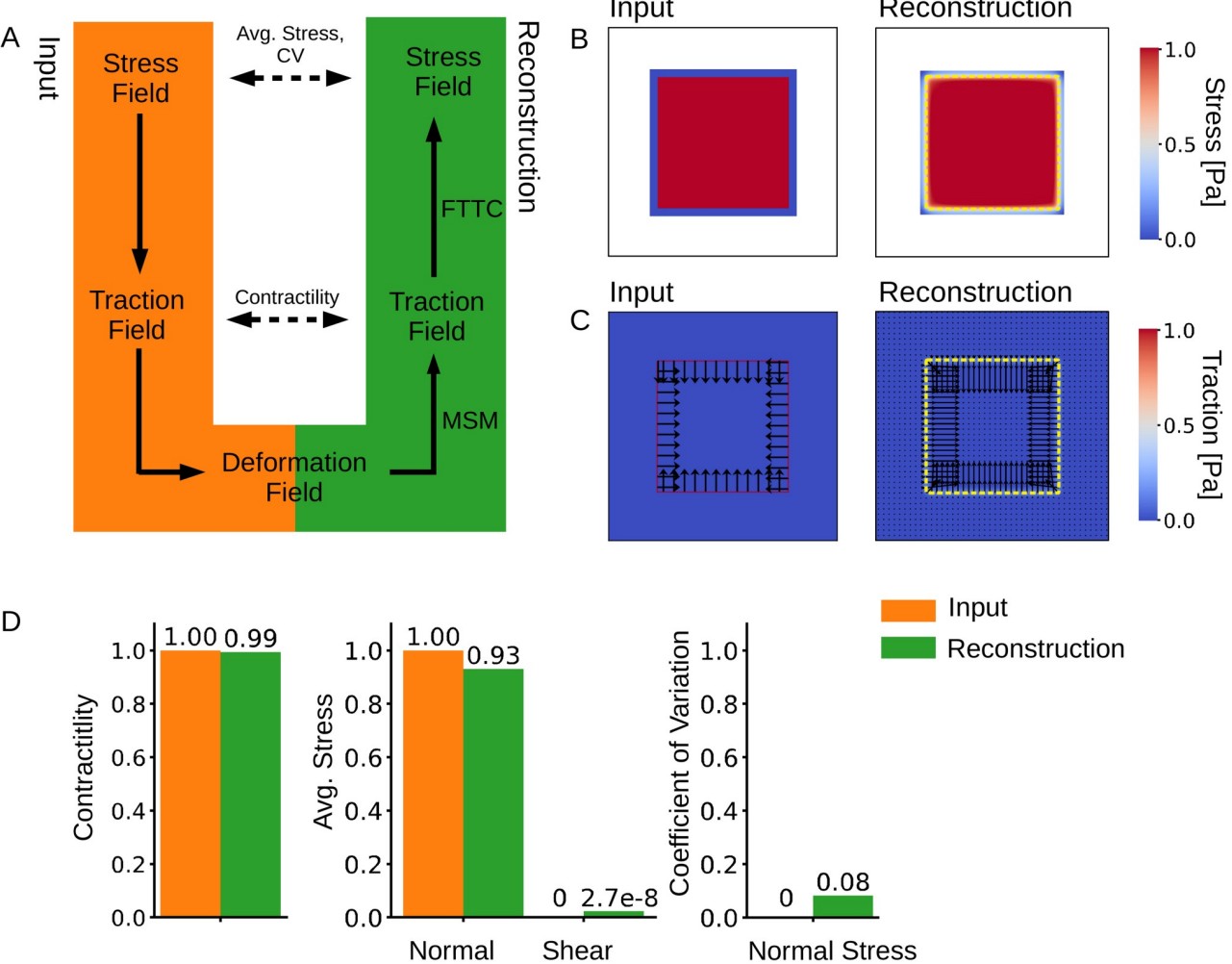

**Fig 3. Accuracy of stress and traction force calculation.** A: We model a cell colony as a uniformly distributed square-shaped stress field for which we analytically compute a traction field and subsequently a deformation field. We use the deformation field as the input for Traction Force Microscopy and Monolayer Stress Microscopy to recover the traction and the stress fields. B: Input and reconstructed traction field. C: Input and reconstructed stress field. The yellow dashed line shows the extent of the original stress field. D: Contractility and average normal and shear stress and CV for the mean normal stress in the input and reconstructed traction and stress fields. The contractility is computed over an area that is 12 μm larger than the original stress field. Average normal and shear stresses and the CV of the mean normal stress are computed over the area of the original stress field.

We find that the pyTFM algorithm accurately reconstructs the stress field (Fig 3B). By contrast, the reconstructed traction field is blurred in comparison to the input traction field (Fig 3C). This is the effect of a Gaussian smoothing filter with a sigma of 3 μm that is applied to the tractions computed by the FTTC algorithm. This filter helps to prevent unphysiological isolated and locally diverging tractions in the case of a noisy input deformation field. In our test case, we do not model the influence of noise and could therefore omit the filter; in practical applications, we find a sigma of 3 μm to give the best compromise between resolution and noise.

The computed average normal stress is slightly (7%) smaller than the input stress, but the error increases rapidly when the margin for extending the FEM-grid is decreased below 5 μm (Fig 4A). Total contractility and the coefficient of variation for the normal stress are recovered accurately (Fig 3D).

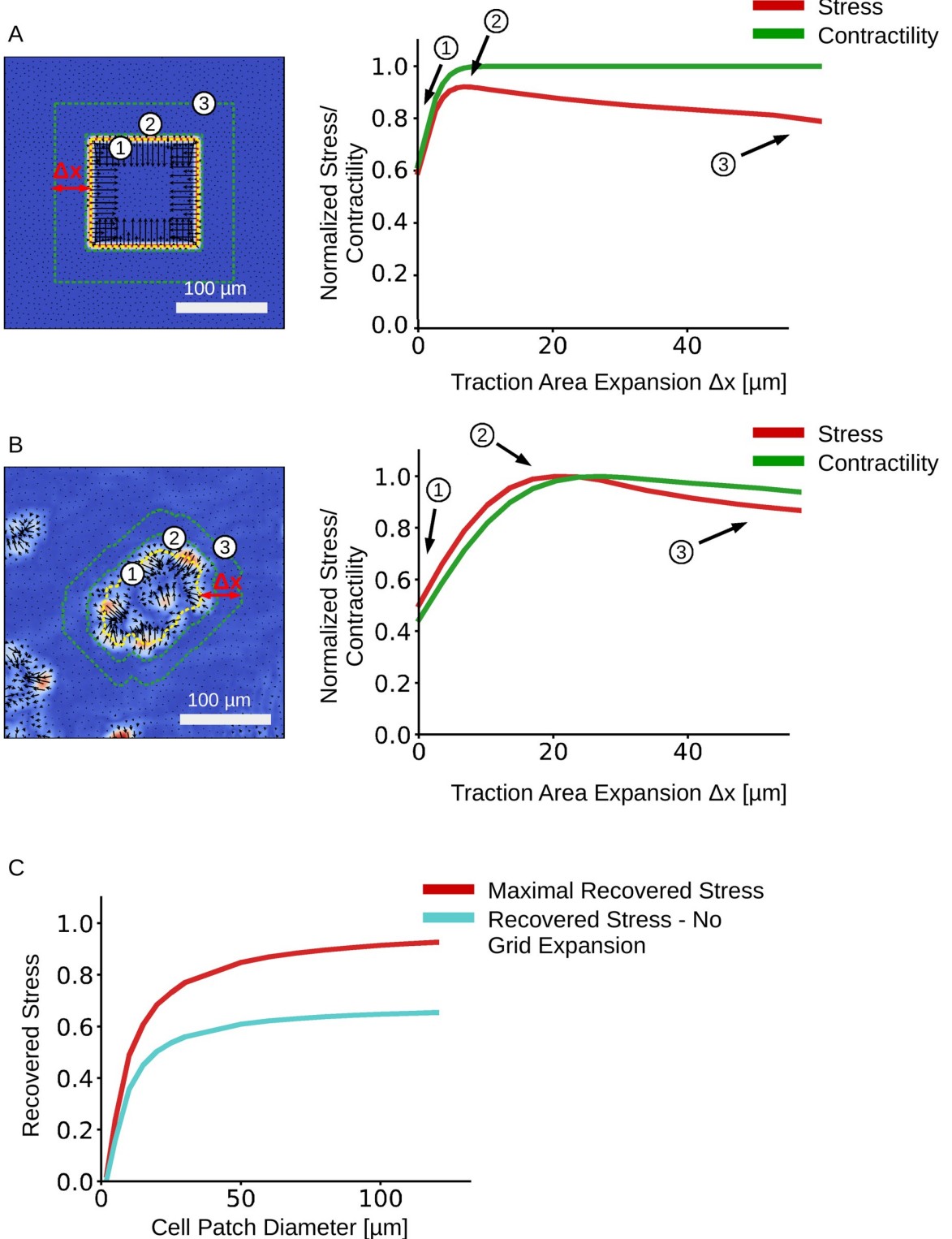

**Fig 4. Effect of increasing the traction area on stress and contractility recovery.** The predicted traction fields of a synthetic test system (A) and an MDCK cell patch (B). The outlines of 3 representative FEM-grids are shown on the left. The relationship between average normal stress and FEM-grid area is shown on the right. C: Influence of the cell patch size of a synthetic data set on the maximally recovered mean normal stress with FEM-grid expansion (red), and the recovered mean normal stress without FEM-grid expansion (turquoise).

**3.1.1 Effect of FEM-grid area on stress recovery.** pyTFM requires the user to select an area of the traction field over which pyTFM then computes contractility and strain energy and draws the FEM-grid for computing monolayer stresses. The size of this area influences the accuracy of the stress and force measurements. Selecting an area that is too small leads to an underestimation of stresses and contractility. Selecting an area that is too large also leads to an underestimation of stresses. To systematically analyze which effect the size of the user-selected area has on the traction and stress reconstruction, we expand the traction area and analyze the average normal stress and the contractility for the synthetic test data described above (Fig 4A) and for an MDCK cell patch grown on a polyacrylamide substrate (Young's modulus 49 kPa, Fig 4B). In the case of the synthetic data, we normalize the computed average normal stress and contractility to the known input stress (1 N/m) and to the known contractility of the input traction field (600 N), respectively. In the case of the experimental data, we normalize the computed average normal stress and contractility to their respective maximum values as the true stress and contractility is unknown.

The influence of FEM-grid area on the stress calculation may also depend on the size of the cell patch. To analyze this effect, we reduce the cell patch size in the synthetic test data (Fig 4A) and calculate the average mean normal stress (Fig 4C). The margin of expansion for each cell patch is chosen so that we recover the maximal stress (the optimal margin of expansion $\Delta x$ is between 4-5 μm, independent of patch size, as it is mainly determined by the spatial accuracy of the TFM algorithm).

We find that the normalized stress rapidly increases (by approximately 40%) with increasing area until it reaches a maximum, after which it declines at a slower rate. The contractility displays a similar initial increase but then remains approximately constant. The maximum of the normalized stress occurs when the traction area just covers all cell-generated tractions, including those that appear outside the cell patch. In the cases of the synthetic data, the maximum is reached at a traction area expansion distance of 5 μm beyond the cell patch outline, whereas in the case of the MDCK cell patch, it is reached at at an expansion distance of 20 μm. The reason for this larger distance in the MDCK data is the additional blurring of tractions introduced by the PIV algorithm (whereas no PIV was needed for analyzing the synthetic data). The traction area corresponding to the maximal normal stress can be regarded as the optimum, as approximately 93% of the input stress is recovered. Expanding the traction area and thus the FEM-grid beyond the optimum distance adds elastic material to the monolayer and thereby reduces the average stress. This stress reduction, however, occurs only gradually (Fig 4B), which implies that in practice it is best to choose the traction area rather generously to include all cell-generated tractions. The contractility reaches its maximum values at almost the same expansion distance as the stress. Thus, it is possible to use the same area to accurately compute both stress and contractility.

The stress recovery also depends on the cell patch sizes (Fig 4C). The recovered mean normal stress increases rapidly with patch size but reaches a plateau above 50 μm so that more that 90% of the input stress is recovered. Expanding the grid area by 4-5 μm, as noted above, always results in a higher stress recovery compared to an FEM-grid area that matches the patch area (Fig 4c red vs. turquoise curves).

## 3.2 Analysis of a MDCK cell-colony with pyTFM

In the following, we illustrate the workflow of pyTFM (Fig 1) using a MDCK cell colony as a representative example. Experimental details for this example are provided in the Supplementary S2 Text. Two images of fluorescent beads serve as the essential input, one image taken before and one image after cell removal by trypsinization of the cells (Fig 5A). pyTFM

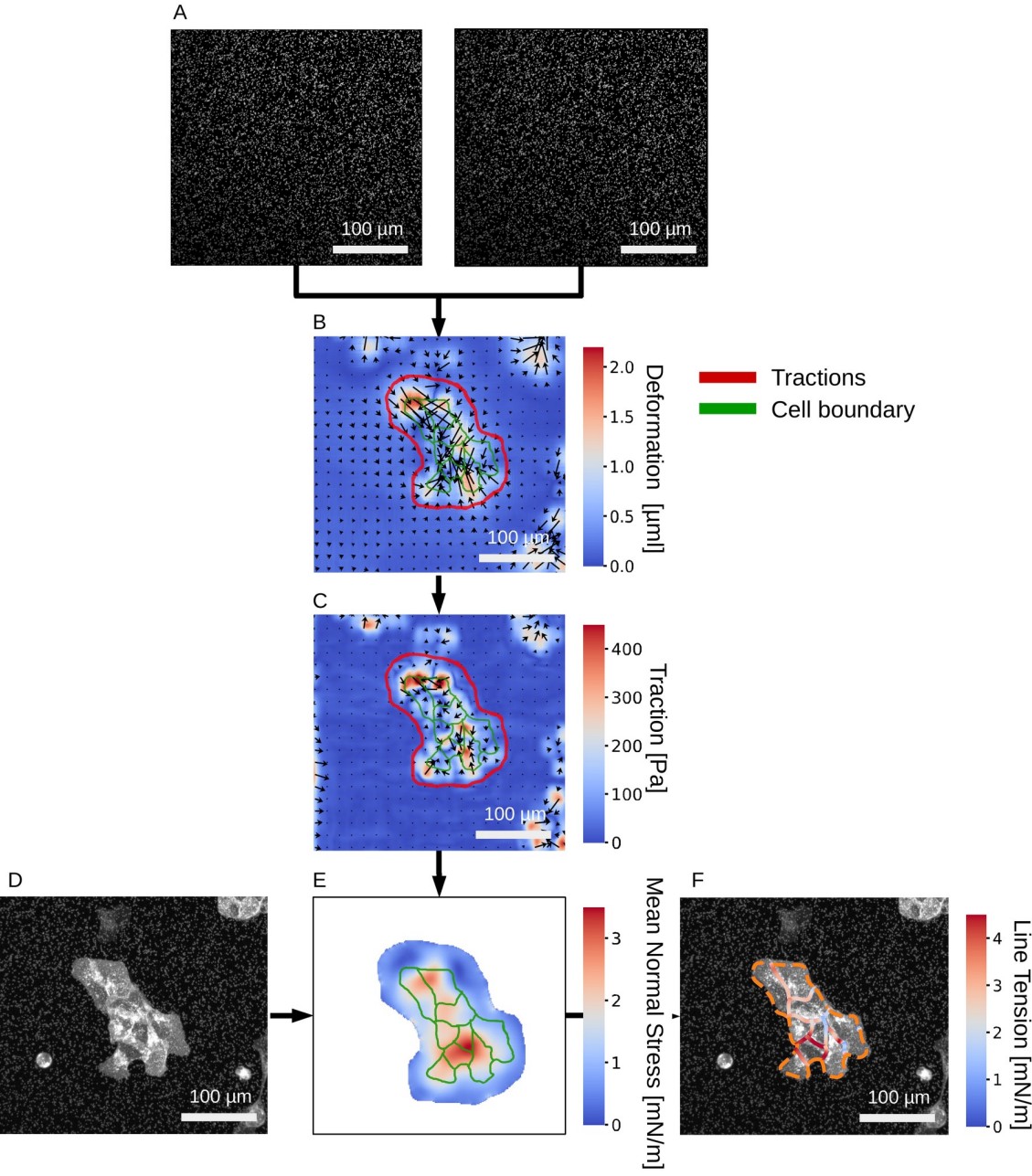

**Fig 5. Analysis of stress and force generation of a MDCK cell colony.** A: Images of substrate-embedded fluorescent beads before and after the cells are detached by trypsinization. B: Substrate deformation field. C: Traction field. The user selects the area (red outline) over which contractility, strain energy and cell stresses are subsequently calculated. D: Image of the cell colony; fluorescent membrane staining with tdTomato-Farnesyl. E: Absolute value of the Mean normal stress in the cell colony. F: Line tension along cell-cell borders. The orange dashed line marks the outer edge of the cell colony.

calculates the deformation field (Fig 5B) and the traction field (Fig 5C). The user then selects the area (red outline in Fig 5) over which pyTFM draws the FEM-grid and computes the contractility and strain energy (both are scalar values), and the monolayer stress field (represented as a map of normal stresses (Fig 5E)). If the user optionally selects the outline of the cell patch and the boundaries of the individual cells within the patch (green outlines in Fig 5E), pyTFM

**Table 1. Scalar values computed by pyTFM quantifying cellular force generation and stress distribution.**

| Scalar Quantity | Result |
| --- | --- |
| Contractility | 0.64 μN |
| Strain energy | 0.11 pJ |
| Avg. max. normal stress | 2.62 mN/m |
| Avg. max. shear stress | 0.78 mN/m |
| CV normal stress | 0.38 |
| Avg. line tension | 2.04 mN/m |
| Avg. normal line tension | 1.94 mN/m |
| Avg. shear line tension | 0.56 mN/m |

also computes the line tension between the cells (Fig 5F). The program also computes a number of scalar values for quantifying cellular force generation and stress distribution (Table 1).

The cell colony in this example displays several typical features: First, stresses and traction forces are unevenly distributed across the cell colony, as indicated for example by a high coefficient of variation of 0.38 for the normal component of the stress field (Table 1). Second, the average line tension is higher than the average normal or maximum shear stress. This indicates that, on average, interfacial stresses between cells exceed intracellular stresses. Third, normal and tensile components of the stress field dominate over shear stress components, indicating that tractions are locally aligned. In addition, the shear component of the line tension is considerably smaller than its normal component, implying that cells in this colony pull on each other but do not exert appreciable forces parallel to their boundaries.

## 4 Availability and future directions

pyTFM provides a user-friendly implementation of Traction Force Microscopy and Monolayer Stress Microscopy in a combined image and data analysis pipeline. For users interested only in Traction Force Microscopy, several other intuitive software packages are freely available:

The FTTC and PIV ImageJ plugins [37], hosted at https://sites.google.com/site/qingzongtseng/tfm#publications, can analyze the typical traction force experiment in which the substrate is imaged once before and once after force relaxation. The software makes use of the ImageJ framework to organize input images and output plots. It calculates deformation fields using the standard PIV algorithm and traction fields using the L2-regularized FTTC algorithm. The deformation field can be filtered with a number of methods. Additionally, this plugin can calculate the strain energy over a user-selected area.

The TFM MATLAB package [28], hosted at https://github.com/DanuserLab/TFM, uses PIV or PTV to calculate the deformation field. Traction fields can be calculated either with the L1- or L2-regularized FTTC algorithm or the L2-regularized Boundary Elements Method. Appropriate regularization parameters can be selected using the L-curve method [28]. This package also allows for the analysis of TFM-experiments where the evolution of cellular forces is measured over time.

Another MATLAB tool [38], hosted at https://data.mendeley.com/datasets/229bnpp8rb/1, implements Bayesian FTTC [12], thus providing a method to automatically select the regularization parameter. Additionally, this package can also perform traditional L2-regularized FTTC and enables the user to manually select the regularization parameter using the L-curve method. However, the user needs to provide the deformation field as an input.

Currently, pyTFM exclusively uses the Fourier Transform Traction Cytometry algorithm [13]. This algorithm is simple, robust and well established but has a number of limitations (see Section 2.2.3). However, due to the structure of pyTFM, it is possible to implement alternative algorithms that address these issues with minimal changes to other parts of the software. An example is the Boundary Elements Method [11] that solves the inverse problem numerically in real space and allows users to set spatial constraints on the tractions. This avoids the occurrence of arguably unphysiological tractions outside the cell area. Another example is 2.5-dimensional Traction Force Microscopy that allows for the calculation of tractions in z-directions [10]. This algorithm is also necessary when cells are grown on compressible substrates and generate significant z-tractions. Finally, FEM-based Traction Force Microscopy algorithms allow for the analysis of cells grown on non-linear elastic substrates such as collagen [39].

pyTFM can be downloaded and installed from https://github.com/fabrylab/pyTFM under the GNU General Public License v3.0. Detailed instructions on the installation and usage are provided at https://pytfm.readthedocs.io/.

## Supporting information

**S1 Text. Scalar quantities used to describe cellular stresses and force generation and FEM-element size on the accuracy of stress calculation.** First, we discuss the definition and interpretation of the quantities that pyTFM uses to describe cellular stresses, force generation and cell-cell force transfer. Next, we analyze the effect of FEM-element size on the stress calculation.
(PDF)

**S2 Text. Experimental details for analyzing the MDCK cell colony.** We provide basic information on our protocols for polyacrylamide gel preparation and cell culture for the TFM analysis of the MDCK cell colony.
(PDF)

**S1 Dataset. Numerical data shown in Fig 4: "Effect of increasing the traction area on stress and contractility recovery".** We provide the exact data that is displayed in the Graphs shown in Fig 4A–4C.
(XLSX)

**S1 Archive. pyTFM source code and documentation.** This archive contains the pyTFM source code and documentation which includes installation and usage instructions and links to further example data sets.
(ZIP)

## Acknowledgments

We thank Richard Gerum for our close collaboration and adapting the image annotation tool ClickPoints to fit our specific needs. Further we thank Christoph Mark for valuable discussion.

## Author Contributions

**Conceptualization:** Ben Fabry.

**Investigation:** Andreas Bauer, Magdalena Prechová, Lena Fischer.

**Methodology:** Andreas Bauer, Lena Fischer.

**Project administration:** Martin Gregor, Ben Fabry.

**Software:** Andreas Bauer.

**Supervision:** Ingo Thievessen, Martin Gregor, Ben Fabry.

**Visualization:** Andreas Bauer.

**Writing – original draft:** Andreas Bauer.

**Writing – review & editing:** Magdalena Prechová, Ben Fabry.

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
