## [Decision Letter · Decision Letter 0]

15 Nov 2020

Dear Mr. Bauer,

Thank you very much for submitting your manuscript "pyTFM: A tool for Traction Force and Monolayer Stress Microscopy" for consideration at PLOS Computational Biology.

As with all papers reviewed by the journal, your manuscript was reviewed by members of the editorial board and by several independent reviewers. In light of the reviews (below this email), we would like to invite the resubmission of a significantly-revised version that takes into account the reviewers' comments.

We cannot make any decision about publication until we have seen the revised manuscript and your response to the reviewers' comments. Your revised manuscript is also likely to be sent to reviewers for further evaluation.

Sincerely,

Manja Marz

Software Editor

PLOS Computational Biology

Reviewer's Responses to Questions

**Comments to the Authors:**

Reviewer #1: Uploaded

Reviewer #2: This is a straightforward, well-written, and highly useful description of a computational approach to traction microscopy and monolayer stress microscopy. These methods allow for the visualization and quantification of stress fields exerted by, upon, and between adherent cells in isolation or in a continuous multicellular collective. Traction microscopy and monolayer stress microscopy have come into wide use in many specialized laboratories across the world, but they currently require a good deal of mathematical/computational expertise that is not generally available. For many laboratories, however, this required expertise serves as an obstacle that in many cases is insurmountable.

This submission makes freely available the computational wherewithal and technical guidance for these methods to become widely available to the non-specialist. While there is nothing new from a scientific point of view, this paper represents a great advance from a practical point of view. As such, I offer my strong recommendation in favor of publication.

Minor:

Page 2, Abstract 3 lines from bottom. This is redundant.

**Have all data underlying the figures and results presented in the manuscript been provided?**

Reviewer #1: Yes

Reviewer #2: Yes

PLOS authors have the option to publish the peer review history of their article (what does this mean?). If published, this will include your full peer review and any attached files.

Reviewer #1: No

Reviewer #2: No
---

## [Decision Letter · Decision Letter 1]

2 Feb 2021

Dear Mr. Bauer,

Thank you very much for submitting your manuscript "pyTFM: A tool for Traction Force and Monolayer Stress Microscopy" for consideration at PLOS Computational Biology.

As with all papers reviewed by the journal, your manuscript was reviewed by members of the editorial board and by several independent reviewers. In light of the reviews (below this email), we would like to invite the resubmission of a significantly-revised version that takes into account the reviewers' comments.

We cannot make any decision about publication until we have seen the revised manuscript and your response to the reviewers' comments. Your revised manuscript is also likely to be sent to reviewers for further evaluation.

Sincerely,

Manja Marz

Software Editor

PLOS Computational Biology

Manja Marz

Software Editor

PLOS Computational Biology

Reviewer's Responses to Questions

**Comments to the Authors:**

Reviewer #1: Authors have addressed all points except one. This issue is in the difference between analysis of a cell patch versus analysis of a small portion of the patch. Not only is this point not appropriately addressed, but the updated text has introduced new problems. These issues are limited to lines 204-213 of the revised manuscript. Some of these issues are minor. Others are likely just miscommunication, but the clearest meaning I can gather from these sentences indicate major problems. These apparently major problems must be addressed before this manuscript can be accepted.

Here are all the remaining problems:

1. “Equation 4 is not uniquely solvable …” - As written the sentence is misleading. The equation is solvable. It is inaccuracies in the measurement of displacement that cause problems. Hence, the sentence must start with the qualifier that “In case of a cell patch, measured displacements can have a nonzero mean translation and rotation. In such case, ”

2. Updated text does not acknowledge the fact that such analysis of a cell patch has been conducted in reference #17.

3. “First, to ensure … corrected by subtracting the net force by rotating all force vectors to enforce zero torque” - the meaning of the last part of the sentence is unclear. What is subtracted?

4. One cannot apply both force and displacement constraint on a node. The assertion “Second, zero …” is likely misstated.

Reviewer #2: Well done.

Reviewer #3: The manuscript by Ben Fabry and colleagues describes a Python package for 2D traction force microscopy and monolayer stress microscopy. The package bundles well established algorithms to perform these computations based on bead-tracked substrate deformations.

To make this review transparent I disclose my identity from the get-go: Gaudenz Danuser.

I am a bit confused as to why I was invited to review this revision of the manuscript beginning of 2021, given that the cover letter to the revision is dated with March 2020. I was not part of the first round of reviews. The authors seem to have responded adequately to the questions the initial pair of reviewers raised. That said, I feel there could have been more input with the first round of reviews, e.g. around the choice of the FTTC approach to traction reconstruction. In my opinion – and I am not alone – this algorithm has significant limitations associated with an implicit regularization. This outweighs the speed advantages. Especially for high-resolution TFM, where subcellular force distributions are analyzed, this approach is very problematic. As the authors certainly are aware, the regularization is the primary Achilles heel of inverse problem solutions, whether they exist in Fourier or direct space. The cell biology/biophysics literature is cluttered with publications of traction force maps, where authors have chosen arbitrary regularization parameters; I am afraid in numerous cases the authors probably did not even know that they were regularizing the traction field. We discussed this problem in fairly extensive detail in Han et al. Nature Methods, 2015 as a preamble to introducing L1-norm regularization for TFM. For whatever reason, this work is not cited by the authors.

While I am highly sympathetic to the authors’ intention of publishing a user-friendly platform that works, I feel the discussion of inverse problem solvers and the fact that regularization is necessary for this step (around lines 26 – 30) should be deeper and more didactical. I am quite concerned when I imagine yet another graduate student without any basis in mechanics turning the knobs of a user-friendly software package. As biophysicists, we should make an effort to avoid this. This applies especially to a publication that targets the introduction of a tool rather than presenting a novel method. I strongly encourage the authors to expand this part and really go to the grounds of pros and cons of the numerical methods implemented in pyTFM, especially since there are other options of software packages with alternative methods implemented (see next paragraph). The Sabass et al. 2008 paper as a reference to such an overview of methods is outdated.

I am also fully supportive of the notion that robust, user-friendly software packages are key to progress in life science experiments. I am not a cognoscenti of the traction force software market. The authors may be correct in saying that pyTFM is advancing that market, especially because they combine traction reconstruction and intercellular stress in the same package. I am not aware of a package that can do that. But, as with the methods, I do not think their rationale for such a package does justice to the state-of-the-art. To give the example I know, the Han et al. paper comes arguably with a pretty user-friendly interface. Accordingly, the package is downloaded reasonably frequently (https://github.com/DanuserLab?tab=repositories) by the same community the authors target; and I just notice that Benedikt Sabass published a paper in 2020 with the title ‘A Bayesian traction force microscopy method with automated denoising in a user-friendly software package’. To make their contribution useful, the present manuscript should be amended by a table that lists other packages and the methods they implement. Ideally, it will delineate which problems are solved best with the present package vs the advantages of other projects for other problems.

With these two important additions in place, I think this is a nice and useful contribution to the field. I hope the authors will have the resources to support and expand the software.

**Have all data underlying the figures and results presented in the manuscript been provided?**

Reviewer #1: Yes

Reviewer #2: Yes

Reviewer #3: **No: **

PLOS authors have the option to publish the peer review history of their article (what does this mean?). If published, this will include your full peer review and any attached files.

Reviewer #1: No

Reviewer #2: No

Reviewer #3: **Yes: **Gaudenz Danuser
---

## [Decision Letter · Decision Letter 2]

2 May 2021

Dear Mr. Bauer,

We are pleased to inform you that your manuscript 'pyTFM: A tool for Traction Force and Monolayer Stress Microscopy' has been provisionally accepted for publication in PLOS Computational Biology.

Best regards,

Manja Marz

Software Editor

PLOS Computational Biology

Manja Marz

Software Editor

PLOS Computational Biology

Reviewer's Responses to Questions

**Comments to the Authors:**

Reviewer #1: Outstanding contribution to cell mechanics

Reviewer #2: The comments by Dr. Danuser are noteworthy, but do not detract appreciably from my enthusiasm for the work.

Reviewer #4: Bauer and coworkers report a new software , pyTFM, to analyze 2D Traction Force Microscopy and Monolayer Stress Microscopy experiments. Compared to other solutions, this software is open source (GPLv3), user friendly and built on top of well mature open source tools such as OpenPIV and SolidsPy. Not only they provide the manuscript and the source code for pyTFM, but it is accompanied by an excellent help and step-by-step tutorial for its use. As pyTFM stands, it is a great addition for the quantitative biology, Open Source and Python communities. The manuscript is well written and technically sound, with some minor changes.

1.- In the paragraph describing equations (6-8), it is implicit that balance of forces and moments is equivalent to balance of displacements and rotations, but this equivalence happens because the monolayer is assumed as linearly elastic. This might not be the case for other materials, so it would be worth to explicitly mention in this paragraph the assumption of linear elasticity. Users of pyTFM might be inclined to model the monolayer, in the future, with different mechanical models and use its FEM to solve it.

2.- As stated above, the authors build pyTFM on top of excelent Open Source Python packages, but they miss the citation of some of them in the manuscript. The authors must cite, at least, Numpy, Cython, Scipy and Matplotlib:

https://numpy.org/citing-numpy/

https://github.com/cython/cython/wiki/FAQ#how-do-i-cite-cython-in-an-academic-paper

https://www.scipy.org/citing.html

https://matplotlib.org/stable/citing.html

3.- After installing pyTFM in one computer, it was not able to run because it was missing one dependence, "yaml". I understand the installer is not checking for this dependency.

4.- The authors provide two ways of using pyTFM, either integrated with ClickPoints or as standalone Python functions. I was able to run pyTFM as functions in a Python script, but sadly, pyTFM was not working for me through ClickPoints. The install and drift correction was successful, but while trying to calculate the deformation field, the computer ram (32GB and 64GB) would fill up and the program would crash. This happened in two different computers. I understand this could be due to my Linux distribution or package versions, but it should not happen in a software aiming for ease to use and user friendliness. The authors should consider adding a list of operating systems/distributions and package versions where pyTFM is confirmed to work. The advantage of pyTFM is that, on being open source, the users would be able to report problems like this directly to their github, and the authors would be able to fix them.

Regarding the software itself, I would like to provide some suggestions that would help improve it in future releases, although their implementation is NOT needed for acceptance of the manuscript:

1.- The authors should consider uploading stable versions of pyTFM to the Python Package Index (PyPI) to ease the installation procedure.

2.- For the numerical Fourier Transforms, the authors should consider the use of pyFFTW or scipy.fft rather than numpy.fft.

3.- The authors should consider adding the example fields to the same Github repository as pyTFM, instead of a different one.

4.- The tutorial explains how to correct drift of the example images, but the provided images are pre-corrected. The authors should consider providing the drifted images, so the user can see the actual effect of the correction.

**Have all data underlying the figures and results presented in the manuscript been provided?**

Reviewer #1: Yes

Reviewer #2: Yes

PLOS authors have the option to publish the peer review history of their article (what does this mean?). If published, this will include your full peer review and any attached files.

Reviewer #1: No

Reviewer #2: No

Reviewer #4: No

**Have the authors made all data and (if applicable) computational code underlying the findings in their manuscript fully available?**

Reviewer #4: Yes

---

## [Editor Report · Acceptance letter]

16 Jun 2021

PCOMPBIOL-D-20-01714R2 

pyTFM: A tool for Traction Force and Monolayer Stress Microscopy

Dear Dr Bauer,

I am pleased to inform you that your manuscript has been formally accepted for publication in PLOS Computational Biology. Your manuscript is now with our production department and you will be notified of the publication date in due course.

With kind regards,

Zita Barta
